# Research Progress of Polymers/Inorganic Nanocomposite Electrical Insulating Materials

**DOI:** 10.3390/molecules27227867

**Published:** 2022-11-15

**Authors:** Guang Yu, Yujia Cheng, Zhuohua Duan

**Affiliations:** Mechanical and Electrical Engineering Institute, University of Electronic Science and Technology of China, Zhongshan Institute, Zhongshan 528400, China

**Keywords:** polymers, inorganic, nanocomposite dielectric

## Abstract

With the rapid development of power, energy, electronic information, rail transit, and aerospace industries, nanocomposite electrical insulating materials have been begun to be widely used as new materials. Polymer/inorganic nanocomposite dielectric materials possess excellent physical and mechanical properties. In addition, numerous unique properties, such as electricity, thermal, sound, light, and magnetic properties are exhibited by these materials. First, the macroscopic quantum tunneling effect, small-size effect, surface effect, and quantum-size effect of nanoparticles are introduced. There are a few anomalous changes in the physical and chemical properties of the matrix, which are caused by these effects. Second, the interaction mechanism between the nanoparticles and polymer matrix is introduced. These include infiltration adsorption theory, chemical bonding, diffusion theory, electrostatic theory, mechanical connection theory, deformation layer theory, and physical adsorption theory. The mechanism of action of the interface on the dielectric properties of the composites is summarized. These are the interface trap effect, interface barrier effect, and homogenization field strength effect. In addition, different interfacial structure models were used to analyze the specific properties of nanocomposite dielectric materials. Finally, the research status of the dielectric properties of nanocomposite dielectric materials in the electrical insulation field is introduced.

## 1. Introduction

With the rapid development of power, energy, electronic information, rail transit, and aerospace industries, nanocomposite dielectric materials have begun to be widely used as new materials [1]. Research on the structure, performance, and application of these materials is a new frontier in science and technology. Recently, polymer nanocomposites reinforced with a lower volume fraction of nanoceramics and carbon nanotubes have attracted steadily-growing interest due to their peculiar and fascinating properties as well as their unique applications in commercial sectors [2,3]. They also promote social progress. Research on the hierarchical structure and macro-properties of nanocomposite dielectric materials is crucial for scientific research and applications in modern technology [4]. For example, the gold/silver nanoparticles modified by Polyethyleneimine (PEI) possess high electron density, large absorption cross section and special surface plasmon resonance (SPR).. They also possess excellent biocompatibility and chemical stability. They are widely used for drug delivery, medical imaging, genetic transmission, biomolecular detection and tissue engineering. Nanostructuring organic polymers and organic/inorganic hybrid materials and controlling blend morphologies at the molecular level are the prerequisites for modern electronic devices, including biological sensors, light-emitting diodes, memory devices and solar cells [5]. To promote the technological progress of nanocomposite dielectric materials, the scientific conference themed “hierarchical structure and macro-property of Nanocomposite dielectric materials” was convened in Beijing in 2009. In this conference, numerous experts agreed that nanocomposite dielectric research is a trans-specialty subject [6]. This research is progressive and challenging. Theoretically, the traditional dielectric theory would develop into a low-dimensional dielectric system that possesses excellent physical and chemical properties and scale effects [7]. The aim is to establish a relationship between the material microstructure and macro-properties. The core of the research is to investigate the dielectric behavior at the mesoscopic scale, which is between macroscopic and microscopic scales. In addition, in technological applications, nanodielectrics possess unique space-time hierarchical structures, which help to promote the development of the field of electrical insulation. According to statistics, 60% of power system failures originate from electrical insulation issues [8]. Dielectric insulating materials and technologies are the basis of electrical and electronic engineering. Dielectric insulating materials determine the service life of power equipment and electronic devices. In the fields of power transmission, rail transit, navigation technology, and aerospace technology, nanodielectric materials and technology are widely used [9].

Nanotechnology is among the new frontier of research fields in the 21st century. Currently, it is also the predominant technology for scientific development. Nanomaterials are a vital infrastructure in nanotechnology. The geometric size of nanomaterials can be achieved at the nanoscale [10]. In addition, nanomaterials possess special properties. Moreover, the composites were prepared by two-phase or multiphase mixing. The composite properties could not be observed in the original signal materials. Among them, polymer nanocomposites are composites with polymers as the matrix. At least one nanoscale-dispersed phase is scattered in the matrix [11]. Compared with that of traditional composites, the nano effect generated from the nano-dispersed phase results in a strong interface interaction between the nanoparticles and the matrix. Therefore, the dielectric properties of polymer nanocomposites are superior to those of traditional polymer composites [12,13]. At very low nanoparticle loadings, results demonstrate some interesting dielectric behaviors for nanocomposites, and some of the electrical properties are found to be unique and advantageous for use in several existing and potential electrical systems [14]. Nanocomposites show many unique properties; there are numerous perspective applications in catalysis, filtering, light absorption, medicine, magnetic medium and for use as new materials, which promotes the development of their basic study; for example, one approach to increase the low thermal conductivity of polymers is to add nanosized or microsized fillers with a relatively high thermal conductivity [15]. This characteristic can be applied to the research and development of electrical insulating materials.

## 2. Fundamental Effects of Nanoparticles

Nanoparticles possess several unique properties. Therefore, the nanoparticle effect is the main reason for the changes in the physical and chemical properties of the matrix. However, the nanoparticle effect also contains several fundamental effects. These effects are the macroscopic quantum tunneling effect, small size effect, surface effect, and quantum size effect [16].

(1) Macroscopic quantum tunneling effect: the quantum tunneling effect indicates that the probability of microparticles with volatility beyond their energy barrier is zero. When applied to macroscopic quantities, such as microparticle magnetization and magnetic flux in quantum coherent devices, it is known as the macroscopic quantum tunneling effect, which was put forward by Fritz Wolfgang in London [17]. The quantum tunneling effect is predominantly used to explain the reason for nickel nanoparticles maintaining superparamagnetism at low temperatures. In addition, it also explains that the phenomena related to the domain wall motion speed are not related to the temperature in the Fe–Ni films. The I(V) characteristic curve is shown in Figure 1.

From Figure 1, the tunneling effect is caused by electrons waving. According to the principles of Quantum Mechanics, at lower speeds, the wavelength of electrons possessing the kinetic energy can be expressed by Formula (1).
(1)$$h\lambda =\sqrt{2}mE$$

In Formula (1), *h* is the Planck constant. *m* is the electron mass. *E* is the kinetic energy of electrons. Moreover, *V* is the barrier potential. If *E > V*, when the electrons get into the barrier potential, their wavelength will change, which can be shown in Formula (2).
(2)$$h\lambda =\sqrt{2}m(E-V)$$

If *E < V*, although the certainly wavelength of waving cannot form, the electrons still get into a certain depth of barrier potential. When the barrier potential is narrow, there are some electrons passing through the barrier potential and the kinetic energy is unchanged. In other words, when *E < V*, the reflected electronic waves must exist because of the electron incident barrier potential. Moreover, the transmitted wave still exists [18].

(2) Quantum size effect: when the particle size decreases to the nanoscale, according to the energy band theory, the electron quasicontinuum level changes to a discrete energy level around the metal nanoparticle Fermi level. In semiconductor nanoparticles, the highest discontinuous molecule orbital energy is occupied, but the lowest molecular orbital energy is unoccupied, and the band gaps increase. According to the electronic models, the level spacing of metallic nanocrystalline could be calculated [19], which is shown in Formula (3).
(3)δ=4Ef/3N

In Formula (3), *Ef* is the fermi potential energy. *N* is the total electrons number. According to this formula, the mean spacing of energy level is in reverse proportion to the total number of free electrons.

(3) Relaxation effect: when the well-dispersed nanoparticles are added into the polymers, according to the low frequency modulus platform, the average displacement between the nanoparticles is equal to particles nearest neighbor spacing. It illustrates the size of particles size is determined by particles concentration. When the cage size is smaller than the particles size, the particles relaxation slow down significantly. To the nanoparticles which are easy to agglomerate in polymers, the size of the particles cage is related to the average size of nanoparticles aggregates. Because the cage effect of nanoparticles exists, the particles escaping from the cage must overcome a certain energy barrier. Moreover, the cages size is smaller, the potential barrier is higher [20]. According to the microviscosity comparison of particles movement in particles network and high polymers entanglement network, the energy barrier is a linear function, which is the ratio of cage size to particles size.

(4) Maxwell-Wagner-Sillars (MWS) effect: if the dielectric is composed by inhomogeneous materials, under the external electric field, the free carrier is captured by the traps and interface during the carrier macro-mobility process. Therefore, the space charges accumulate in the area of interface [21]. The electric dipole moment forms in the areas of charge distribution non-uniformly. This is described as the MWS effect.

(5) Thermal activation energy: because of the thermal activation energy, the atoms gain enough energy and jump into the void. This void is occupied. At this moment, one void forms in original location of this atom. This process can be seen as the voids’ migration to the position of adjacent lattice points. The nanoparticles are smaller, their rotation is more active. In the effect of thermal activation energy, the nanoparticles move flexibly in composites [22].

Nanocomposite interface bonding theory: the polymer/inorganic nanocomposite interface is the interaction region between the inorganic particles and the polymer matrix. From the test results, the comprehensive performance of composites is not simply the cumulation of single-component performances. Each component plays an independent role, and depends on each other within the interface. The interface interaction of polymers/inorganic nanocomposites is mainly through covalent bonds, hydrogen bonds, van der Waals forces, and electrostatic interactions. The mechanism of the action between the two phases can be summarized as the infiltration adsorption, chemical bonding, diffusion, electrostatic, mechanical connection, deformable layer, and physical adsorption theories.

Infiltration adsorption theory: in this theory, the high polymer adhesion can be divided into two phases: first, the high polymer macromolecular chains approach the polar group on the adhered surface according to the macro-and micro-Brownian movement. The nanofluid viscosity is higher with the nanoparticles increasing. This is because the frictional resistance between the particles in nanofluid increases with the nanoparticles increasing. For the same content, the smaller the nanoparticles size is, the higher the nanofluid viscosity is. The nanofluid viscosity is lower with the increasing temperature. The main reason is the temperature rising causes the molecules distance to increase and the attraction between molecules decreases. Second, when the distance between the adherend and the binder molecule is less than 0.5 nm, the van der Waals force produces adsorption.

Chemical bond: when the groups in the two substances react, the two substances join via chemical bonding, and an interfacial structure is formed. The strength of interfacial bonding depends on the quantity and type of chemical bonds between the two substances. Because of the high energy in chemical bonds, these interfaces are stable. A prime example is the interface formed between the fiberglass and the coupling agent in the matrix. The coupling agent is a layer of material with a bifunctional coating on the glass fiber surface. Parts of the functional group in the coupling agent form covalent bonds with the molecules on the glass fiber surface. Other parts of the functional group connect with the matrix resin via chemical bonding. Both combine closely with the coupling agent.

Diffusion theory: the head end of the macromolecular segment and extending end of branched chains on two high polymer surfaces will cause diffusion and entanglement on the interaction surface, thereby forming a molecular network. The degree of diffusion and entanglement is related to the molecular structure, component, and free energy of the molecular segments on the matrix surface. After surface treatment, the surface molecules are effectively diffused and entangled in the interface action area. The degree of closeness of the interface depends on the head end of the macromolecular segment quantity, molecular polarity, interactive force, and entanglement molecule quantity. In this theory, a combination of chemical bonds with molecular diffusion forms an interpenetrating polymer network structure at a certain thickness of the interface layer.

Electrostatic theory: contact between two substances causes the formation of an electric double layer, from which the electrostatic interactions form. In this theory, it is challenging to form a close interface structure using polymers of similar polarities. However, a strong bond strength exists between homopolar polymers, which forms a close interface structure.

Mechanical connection theory: in this theory, the bonding between two phases is realized only via mechanical action. Therefore, mechanical connection theory matches the bonding theory.

Deformable layer theory: in this theory, after the material surface modification, the treatment agent forms a plastic layer on the interface surface. This can decrease the interfacial stress. In addition, it can reduce the effect of interfacial forces.

Deformable layer theory: in this theory, surface modification is used as a part of the interface. Its mechanical properties play a role in the even transferring of stress and weakening of the interfacial stress. In addition, it plays a buffering role.

Physical adsorption theory: when the adsorption of each component is different, the close interface structure can be constituted by physical adsorption.

Maxwell-Wagner sillar phenomena: as the interfacial polarization, the free charge carrier accumulate in interface under electric field effect due to the difference of dielectric constant and conductivity between the heterogeneous. The polarization generates. In low filler content, the matrix crystalline characteristics in composite make the main effect on Maxwell-Wagner polarization intensity. With a one-dimensional crystal distribution model, the interface density in composites is characterized. Combining with the position of composites loss peak, there is obvious negative correlation between interface density with Maxwell-Wagner interfacial polarization slack time.

In nanocomposites, as the nanoparticle size decreases, the specific surface area of the nanoparticles increases. In the polymer matrix, the proportion of the interface structure also increases. Therefore, the interface structure is the most important factor affecting the properties of composites. The factors influencing interfacial bonding are shown in Figure 2.

As shown in Figure 1, the effect of interfacial bonding on composite properties is generalized according to the nanoparticle surface properties and polymer matrix. In the research of nanodielectrics, the action mechanism of the interface on the dielectric properties of composites can be summarized as follows:

(1) Interface trap effect: adding nanoparticles into the matrix introduces numerous interface traps. These traps effectively capture the carriers. Thus, the carrier mobility and mean free path are reduced, thereby affecting the charge transportation process and conductivity properties of the nanodielectric. The carrier energy in the dielectric decreases. Thus, the destruction caused by the carrier impact the polymer molecular segment decreases, consequently prolonging the polymer service life [23].

(2) Interface barrier potential effect: the nanocomposites possess excellent properties such as partial discharge-resistance aging, electrical tree resistance, electrical leakage mark resistance, and thermal conductivity. This is due to the blocking and scattering effects of the interface barrier between the polymer matrix and the inorganic nanoparticles. In addition, inorganic nanoparticles provide a pathway for phonon thermal conduction [24].

(3) Homogenization field strength effect: the trapped carrier causes homopolar charge accumulation in the local polymer local. A reversed electric field appears, which offsets part of the applied electric field. The field strength required for the charges injected from the electrode increases. On the one hand, space charge accumulation decreases. On the other hand, the internal electric field distribution homogenization of the polymer increases the short-time breakdown voltage of the polymers [25].

In addition, the polymer/inorganic nanoparticle interface has distinct effects on the composite crystallinity, chain entanglement density, and molecular movements [26].

## 3. Interface Model of Nanocomposite Dielectric

The interface structure of a nanodielectric is an important factor that affects the dielectric properties. Therefore, in T. J. Lewis’s opinion, the interface structure is essentially the same as that of the nano-dielectric. Since 2000, researchers have discussed the relationship between the interface and dielectric properties from different angles. Different interface structure models were used to explain the special properties of nanocomposite dielectrics [27]. The main interface structure models currently are as follows.

(1)Wikes model of interface: the different interfaces depend on the nanoparticles or the polymer matrix properties. The morphology of the mesoscopic region can be observed using X-ray diffraction, and the Wikes model of the interface is shown in Figure 3 [28]. The sol-gel method is one kind of stable transparent sols system. It uses the chemical compound with high chemically active component as the precursor. The raw material is mixed uniformly under liquid phase. The material goes through the hydrolysis and condensation. The sol polymerizes slowly during the aged micelle, from which the gel with three-dimensional net structure generates. After the drying and sintering, the nanostructure material is prepared. When the polymers react with nano-SiO_2_ particles via the sol-gel method, a Si-rich layer forms at the interface. The nano-SiO_2_ particles are compatible with polymers, and two states are present:
(a)Some of the nano-SiO_2_ particles spread in the covalent bond system outside the polymer area.(b)The interpenetrating network structure is formed via hydrogen bond interactions.(2)Polymer-binding model: based on the theory of mechanical interaction, in the polymer binding model, the macromolecular chains of the polymer matrix are restricted in the vicinity of the nanofiller interface, and a polymer-binding layer is formed. The thickness of the binding layer depends on the interaction force between the polymer molecular chains and nanofillers. If this interaction force is strong, the thickness of the binding layer increases. The polymer-binding model is shown in Figure 4 [29].

(3)Interface ordering model: the interface thickness between the polymer matrix and nanoparticles is related to the chemical bond or physical bond. The polymer macromolecular chains are attached to the nanoparticle surface via physical and chemical actions. The macromolecular chains are arranged on the nanoparticle surface in radiating or parallel ways. The structure was ordered around the nanoparticles. The interface structure depends on the polymer matrix as well as on the addition of particle types and surface modification conditions. Because the forces acting between the nanoparticles and the polymer matrix are different, the different interfaces have different effects on the nanocomposite properties. When they possess the same polarity, the polymer macromolecular chains connect to the nanoparticles via bonding interactions. An orderly arrangement forms. A sketch map of the interface-ordering model is shown in Figure 5 [30].

(4)Single-layer structure model: this model was proposed by Professor T. J. Lewis at Wales University. It is based on the research of colloid chemistry double electrical layer theory, shown in Figure 6. A shows the addition of the nanoparticles, B shows the polymer matrix, and AB shows the interface between the nanoparticles and polymer matrix. Suppose that the nanoparticles disperse uniformly in the matrix, and the image charges appear in the matrix because of the adsorption from the ionizing in the nanoparticle surface groups or ions in the matrix. Professor T. J. Lewis believes that the electron polarization and permanent dipole steering in the matrix can be expressed by the Born formula, which is shown in Formula (1). In formula 4, *U* is the lattice energy of ionic crystal. *M* is Madelung constant, which is related to ion crystal structure. *Z*_+_ and *Z*_−_ are cationic charges and anionic charges. *r*_0_ is the spacing between the adjacent cation and anion. The unit of *U* is Kj/mol. The unit of *r*_0_ is pm.


(4)
$$U=\frac{138940MZ_{+}Z_{-}}{r_{0}}\left(1-\frac{1}{n}\right)$$


In the matrix, the electrolyzed and movable charges produce a spreading charge dielectric bilayer. It can be defined using the Poisson and Boltzmann equations [31], which are shown in Formulae (5) and (6). In Formula (5), Δ is Laplacian Operator. *f* and *φ* are real-value and complex-value equations in manifold. In Formula (6), *h = 2kTm*. *k* is Boltzmann constant. *ρ*, *v* and *T* are density, velocity and temperature under the equilibrium state.
(5)Δϕ=f
(6)$$f_{0}=\left(\rho /m\right){\left(h/\pi \right)}^{3/2}\exp\left(-hv^{2}\right)$$

(5)Multi-core model: the spherical inorganic nanoparticle/polymer composite multi-core model was proposed by Professor Tanaka at Waseda University. It is based on the single-layer structure model proposed by Professor T. J. Lewis. This model is proposed to describe the interface structure and charge-transport properties [32], as shown in Figure 7. Professor T.J. Lewis finds the nanocomposites show unique heat and electrical properties comparing with the traditional microcomposites. The glass transition temperature changes. The dielectric constant decreases (the nanoparticles adding restrains the interfacial polarization between the particles with the matrix effectively). The space charge is restrained. Therefore, the nanocomposites become the research focus in the field of electrical insulating materials and electronic device.

Professor Tanaka believed that the nanocomposite dielectric interface is a transition region. The interface area consists of bonding layer, binding layer, and loose layer. This model has been accepted by numerous researchers. The multi-core mode was used to qualitatively explain the nano-dielectric property changes in the experiment. These include the breakdown property [33], dielectric constant, partial discharge property, space charge inhibition, electrical tree properties [34], dielectric conductivity, and corona-resistance properties. Moreover, professor Tanaka uses the multi-core mode to analyze the partial discharge-resistance property and electrical tree resistance property of nanodielectrics [35]. He believes that the interface structure between polymers with inorganic particles is the interaction zone, whose size is not larger than 20 nm. Because of the small-size effect of nanoparticles, when the nanoparticles are added into the polymer matrix, the percolation thresholds in the interface change. Furthermore, the dielectric properties of the polymer nanocomposites are affected. Professor Tanaka also explores the trap characteristics of LDPE/MgO nanocomposite. He finds that the nanoparticles adding introduces the 1.5–5 Ev deep trap levels [36]. In some studies, the nanocomposite-free volume exists in the loose layer of the multi-core mode. The addition of nanoparticles to the polymer matrix can effectively reduce the free volume. The composite breakdown field strength value increased [37,38,39].

Doctor Hornak concludes that significant improvements have been achieved in the areas of nanocomposites preparation and further applications (e.g., in industry, agriculture, and medicine). One of these promising materials is magnesium oxide (MgO), the unique properties of which make it a suitable candidate for use in a wide range of applications [40].

With the constant study of nanodielectric properties, combining the interface structure with macro performance analysis would provide a theoretical basis for the dielectric properties of nanocomposites. Although there are several interface models to explain the dielectric properties of nanocomposites, few limitations and defects remain. The current model could not explain every experimental phenomenon. Therefore, the development trend of nanodielectrics should be further studied.

## 4. Research Status of Nanocomposites Dielectric Properties

Polymer/inorganic nanocomposites have been widely used by researchers because of their excellent dielectric properties. Since 2002, researchers have conducted numerous experiments on nanocomposite polarization performance, breakdown properties, partial discharge resistance properties, and space charge properties. There have been some predominant achievements in this regard.

The polyimide/Al_2_Si(PI/AI-SI system) refer to the carbon atom in molecules interconnect by the bonding methods of sp^2^ and sp. The delocalization molecular orbital organic molecules generate, which is shown in Figure 8. Professor Murugaraj explored the polarization performance of polyimide composites with nano-Al_2_O_3_ and nano-SiO_2_ particles. Based on the interface phase regulation, the effect of the nanoparticle volume fraction on the composite dielectric constant was explored. From the test results, in the PI–Al system, the interface phase was expressed by *K*_0_, which was 3.24, and a dielectric constant of 280 was achieved. In the PI–Si system, *K*_0_ was 5.01, and a dielectric constant of 19.24 was achieved. Researchers believe that the high dielectric constant is due to the interfacial phase polarization. The effects of chemical bonds and long-range forces exist between the nanoparticles and polymers. Professor Saha also proposes that interface polarization plays a significant role in the high dielectric constant form [41].

Professor Zhang studied the effect of nano-montmorillonite (MMT) particles on the breakdown properties of low-density polyethylene (LDPE). When the experimental temperature was set to 50–60 °C, the breakdown field strength of the nanocomposite was higher than that of pure LDPE by a factor of 1.1. The breakdown holes of the nanocomposite samples and polyethylene samples were compared using scanning electron microscopy. From the test results, the damage of the electric field on the composites was relatively small, as shown in Figure 9 [42].

Professor Yu tried to add the nano-ZnO particles into crosslinked polystyrene (PS). The scanning electron microscope (TEM) was used to character the pure PS and nano-ZnO/PS. From the test result of TEM, the nanoparticles agglomeration of pure PS was clear enough. The grain size distribution was relatively wide. The nanoparticles agglomeration of nano-ZnO/PS was not obvious. The relative particles morphology and dispersivity were both good. The particles surface was smooth. The shape was regular. The average grain diameter was about 20 nm, as shown in Figure 10 [43].

Professor Guo studied the effect of nano-ZnO and nano-SiC particles on the non-linear conductance of a polyethylene matrix. From the experimental results, these two nanoparticles improved the polyethylene nonlinear coefficient to varying degrees. In addition, the composite electrical tree breakdown time was promoted [44]. The electrical tree growth curves of nano-ZnO/LDPE with different ZnO contents are shown in Figure 11.

Professor Cheng studied the effect of nano-ZnO particles on the breakdown properties of LDPE. As observed from the experimental results, when the mass fraction of nano-ZnO particles was 3 wt%, the sample breakdown field-strength value was the highest, which is shown in Figure 12. In addition, the addition of nano-ZnO particles plays the role of an ultraviolet ray inhibitor [45].

Professor Thomas conducted the corona resistance test on silicone rubber composites with different mass fractions of nanoparticle. In this test, the withstand voltage durability of insulation material under the surface partial discharge or corona can be measured. A certain voltage is applied in the samples. In condition of surface discharge, the samples are broken down. From the test results, when the mass fraction of nanoparticles was 1–3%, the corona-resistant performance of the nanocomposites was better than that of pure silicon rubber. The widths of the electric cracks in the different samples are shown in Figure 13 [46].

Professor Zhang studied the electrical tree growth properties of nano-MMT/LDPE composites [47]. As shown by the test results, the addition of nano-MMT particles effectively restrained the growth rate of the electrical tree. The fractal dimension of the electrical tree increased, and the diffusion rate decreased. The stagnated period appears earlier, and the duration is long. The effect of nano-MMT particles on the electrical tree growth channel is shown in Figure 14.

Professor Ma studied the effect of adding nano-TiO_2_ particles on the polyethylene space charge performance. As shown by the test results, after the aminoethyl-aminopropyl-dimethylsiloxanesurface modification on nano-TiO_2_ particles, the nanocomposite space charge inhibition was significantly enhanced. In addition, the dielectric strength did not decrease [48].

Professor Tu studied the space charge characteristics of polyethylene composites with a 1% nanoparticle mass fraction. As shown by the test results, the addition of nanoparticles introduces shallow electron traps and electron hole deep traps in the polymer matrix, which change the polyethylene band structure. The Ohmic contact between the dielectric and electrode changes to a blocking contact. Charge injection has an inhibitory effect. Thus, the space charge accumulation in the dielectric decreases [49]. The charges conduction path formation of nano-polymer is shown in Figure 15.

Professor Zha studied the effect of electrical aging on the space charge properties of polyimide/nanoTiO_2_ composites. As shown by the test results, adding nanoparticles introduces the local trap level, which changes the space charge distribution in the polymers. Therefore, the service life of the samples was prolonged [50].

Professor Yin studied the effect of adding nano-MgO particles on the space charge distribution in a polyethylene matrix. From the test results, when the mass fraction of nano-MgO particles was less than 2%, the space charge injection volume of the nanocomposite was lower than that of pure LDPE. When the applied electric field was higher than 60 kV/mm, homopolar space charges appeared in the nanocomposite. As shown by the test results of the conductance current characteristics, adding nano-MgO particles could raise the threshold electric field according to the space charge accumulation in the LDPE matrix. When the nano-MgO particle sizes are 30 and 80 nm, the space charge accumulation in the nano-MgO composites is less. When the nano-MgO particle size is larger than 80 nm or smaller than 30 nm, the space charge distribution condition in nanocomposites is complex, as shown in Figure 16. In addition, the electrode materials have a significant effect on the injection of the sample space charges [51].

Professor Green added nano-MMT particles to LDPE [52]. As shown by the experimental results, the nano-MMT particles after functionalization spread uniformly in LDPE. The lateral scale of nano-MMT particles is at the nanoscale, with a thickness of approximately 10 nm. Adding nano-MMT particles significantly improved the breakdown field strength of LDPE, which is shown in Figure 17. However, the dielectric loss of the nano-MMT/LDPE composites increased.

## 5. Conclusions

Nanoparticles possess a large specific surface area. As the particle size decreased, the number of surface atoms, surface energy, and surface tension increased sharply. Thus, the interface interaction between the nanoparticles and matrix is strong. The physical and mechanical properties of nanocomposites are significantly superior to those of conventional composites with the same components. The nanoparticles also provide thermal, magnetic, and optical characteristics and dimensional stability to the composites. Therefore, nanocomposite preparation is an important method for obtaining high-performance materials. There are three common structural features in all nanomaterials: (1) the structural units are in the nanoscale or characteristic dimension size are in the nanoscale (1–100 nm). (2) There are numerous interfaces and free surfaces. (3) There are either strong or weak interactions between each unit. Because of their special structures, nanomaterials possess some unique characteristics. In addition, when the physical scale reaches the nanoscale (smaller than 100 nm), the material possesses strong electrical neutrality retention. At a low temperature, the free electronic energy level discretization happens. Some properties in the system such as magnetic susceptibility, specific heat capacity and nuclear magnetic resonance are affected (Kubo effect) [53]. The change in the surface (Fermi surface) electron levels promote the nano-materials exhibiting a unique performance. Thus, material modification is relatively easy in nanocomposites, and these materials are widely used. Except the application in electrical insulating materials, many scholars explored the effect of different types of nanoparticles on dielectric response in other polymer materials. The representative research is as follow:

(1) Polymer-stabilized liquid crystal (PSLC) devices are widely used in various smart light modulation occasions. Their electro-optical properties can still be improved to address future challenges. Professor Yan added the nanoparticles into liquid crystal (LC) materials to change the material’s electro-optical performance. PSLC devices with AgNPs doping have lower driving voltage and response time than un-doped PSLC devices [54].

(2) Nanocomposites of magnetic nanoparticles and polymer matrices combine the properties of their components, and as such are good examples of functional nanomaterials with excellent application potential. Professor Frickel directly compare the behavior of polyurethane films doped with superparamagnetic Fe_3_O_4_, and blocked ferromagnetic CoFe_2_O_4_ nanoparticles. The results of dielectric spectroscopy experiments revealed different effects of Fe_3_O_4_ and CoFe_2_O_4_ nanoparticles on polymer dynamics [55].

(3) According to the research of professor Jia, the polymer dispersed liquid crystal (PDLC) films were prepared using a nematic liquid crystal, photo-curable polymer and TiO_2_ nanoparticles by polymerization-induced phase separation (PIPS) method. Electro-optical properties of doped and undoped samples including transmittance, driving voltage, contrast ratio and response time of the transmittance-voltage were measured, compared and analyzed. The morphology of polymer network has been investigated by means of SEM. The phase separation of LC droplets was found to be dependent on the content of prepolymer, LC and nanoparticles used. The high dielectric properties and high refractive index of TiO_2_ nanoparticles make PDLC have low driving voltage and high contrast. In particular, the nanoparticles concentration can be optimized to obtain promising electronic materials with minimum threshold and high contrast for display applications [56].

(4) The influence of different concentrations (0.5, 1.0, and 2.0 wt.%) of Zinc Oxide (ZnO) filler on the dielectric properties of the cold-curing polyurethane (PU) resin is presented in the study of professor Kudelcik. For this purpose, the direct DC conductivity and the broadband dielectric spectroscopy measurements were used to describe the changes in dielectric responses of PU/ZnO nanocomposites over the frequency and temperature range, respectively. The presence of nanoparticles in the polyurethane resin affected the segmental dynamics of the polymer chain and changed a charge distribution in the given system. These changes caused a shift of local relaxation peaks in the spectra of imaginary permittivity and dissipation factor of nanocomposites. It is suggested that the temperature-dependent transition of the electric properties in the nano-composite is closely associated with the alpha-relaxation and intermediate dipolar effects (IDE) [57].

(5) The effect of in situ formed silver nanoparticles doping on electrorheological response of highly porous chitosan particles in suspensions of polydimethylsiloxane (silicone oil) is considered by professor Kuznetsov. Silver nanoparticles are directly reduced by chitosan from solution. Highly porous chitosan particles with different silver content are fabricated by spraying from solution followed by freeze-drying. The nature of the electrorheological effect is considered from the standpoint of dielectric spectroscopy. The activation energies of polarization processes are determined from the temperature dependences of the dielectric loss modulus. The study shows the opportunity to control the properties of stimuli-responsive materials by changing the structure and physicochemical properties of the functional filler. This approach opens up new possibilities for creating materials with high performance and predetermined properties [58].

Aiming at the introduction of nanophase, the following key scientific problems must be emphatically researched:

(1) The organization and control mechanism and properties evolution rules of nano-composites present the obvious multi-scale and multiphysics characteristics. Determining how to control the nanophase morphology, size, distribution and quantitative analysis, with the effect on nanocomposites properties is very difficult. So, the basic research of nanocomposites must be strengthened, from which the mechanism of nanoparticles adding onto polymers could be explored.

(2) The progressive and scientific characterization and testing instruments need to be further improved and developed, from which the nanocomposites properties can be characterized from the micro level. The essence of these excellent properties can be explored.

(3) The multifunctional characteristics of nanocomposites involve many subjects. So, the interdisciplinary and multidisciplinary integration in nanocomposites research must be utilized.

## Figures and Tables

**Figure 1 molecules-27-07867-f001:**
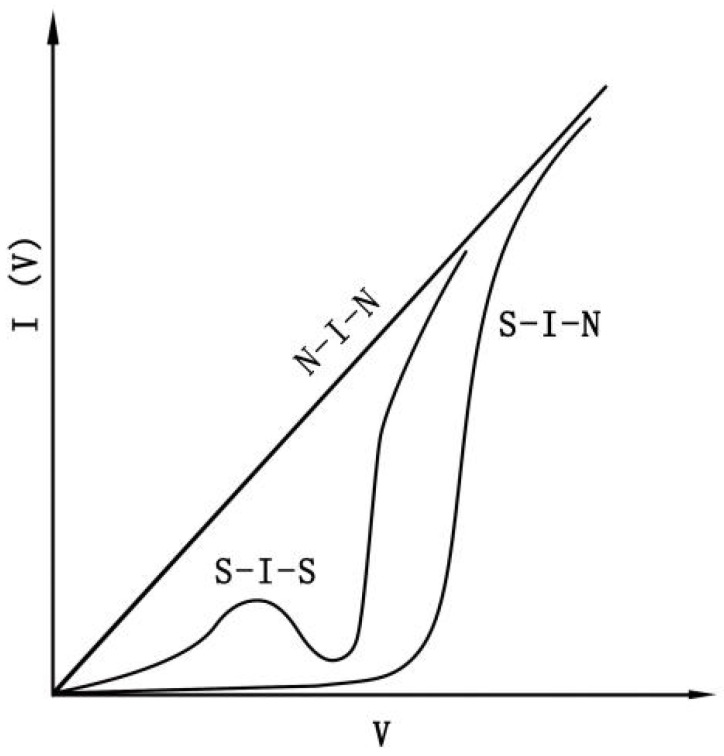
I(V) characteristic curve of tunnel junction.

**Figure 2 molecules-27-07867-f002:**
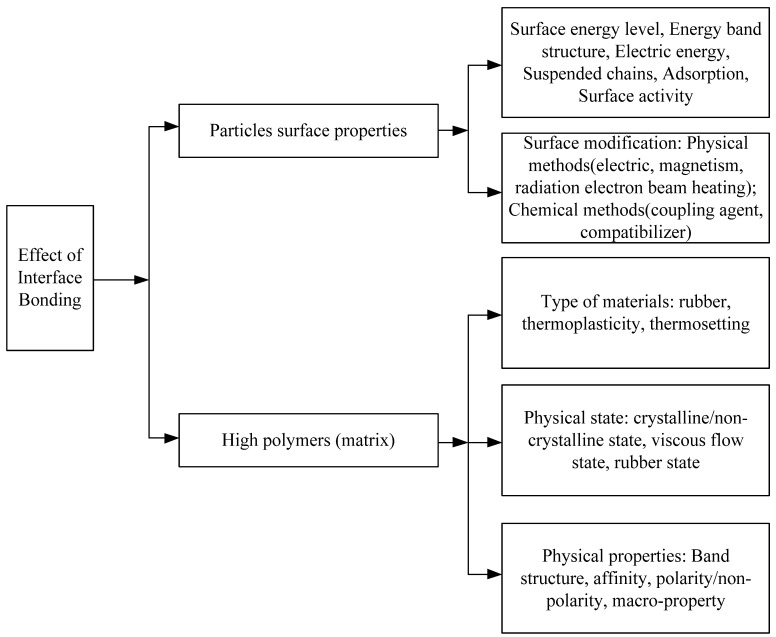
The factors which affect the interface binding.

**Figure 3 molecules-27-07867-f003:**
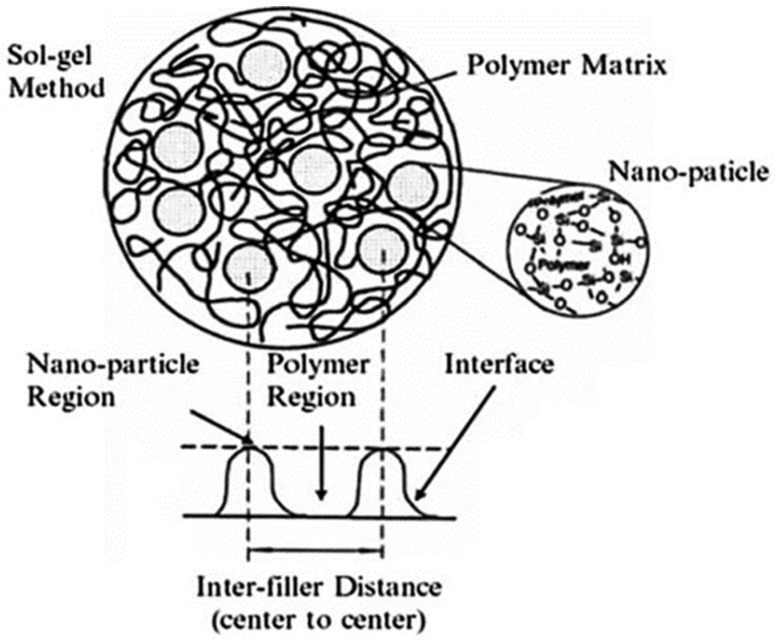
Wikes model of interface.

**Figure 4 molecules-27-07867-f004:**
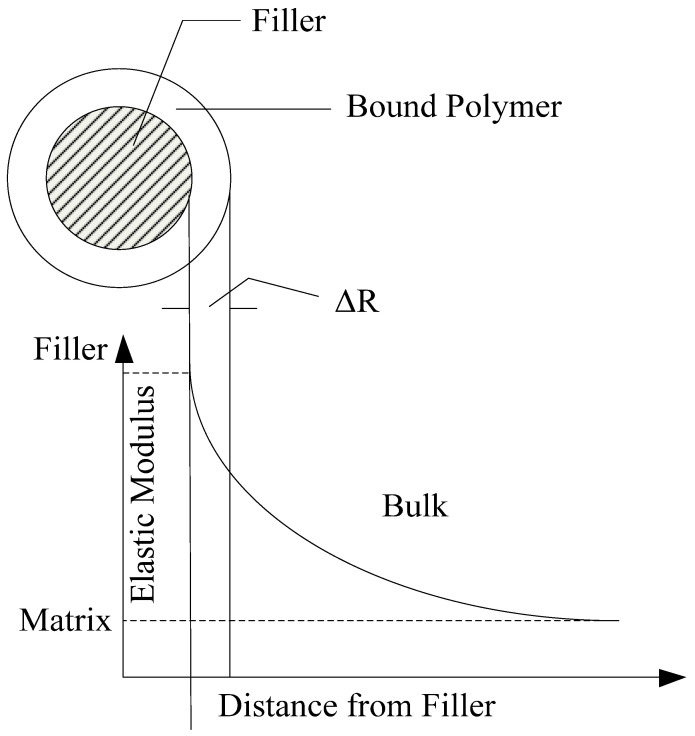
Polymer-binding mode.

**Figure 5 molecules-27-07867-f005:**
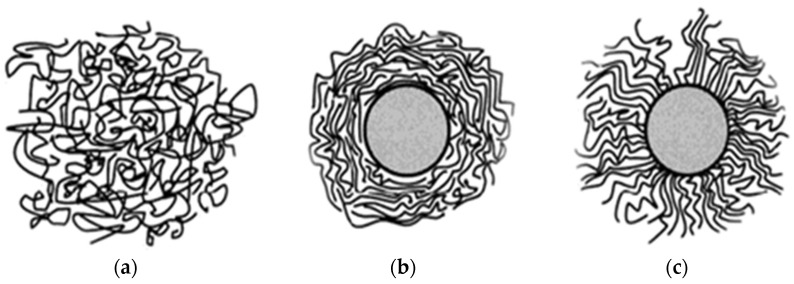
Interface ordering model. (**a**) High polymer chain free arrangement in polymer matrix; (**b**) high polymer chain paralleled arrangement around the nanoparticles; (**c**) radial high polymer chain bonding with nanoparticles.

**Figure 6 molecules-27-07867-f006:**
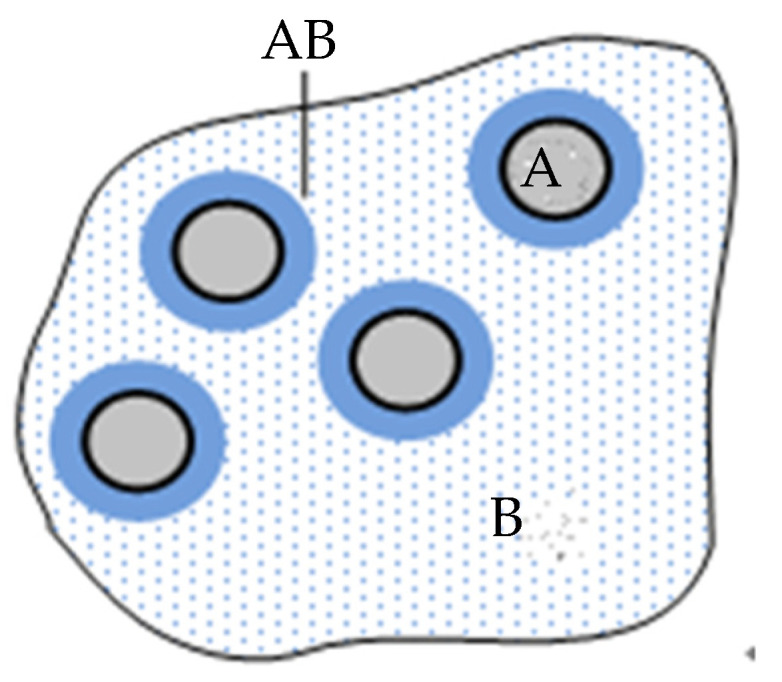
Single-layer structure model.

**Figure 7 molecules-27-07867-f007:**
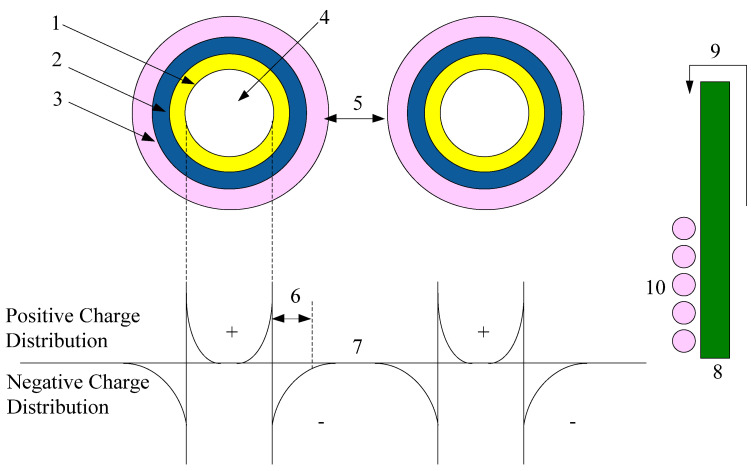
Multi-core model for interpreting several phenomena inherent to nanocomposites. 1, binding layer; 2, trapping layer; 3, loosening layer; 4, nano-particle; 5, interparticle distance; 6, Debye shielding distance; 7, possible overlapping of the third layers and charge tails of nano-particles; 8, electrode facing accumulated charge tails of nano-fillers; 9, charge carrier injection via Schottky emission at high electric field; 10, collective charge tail effect will modify.

**Figure 8 molecules-27-07867-f008:**
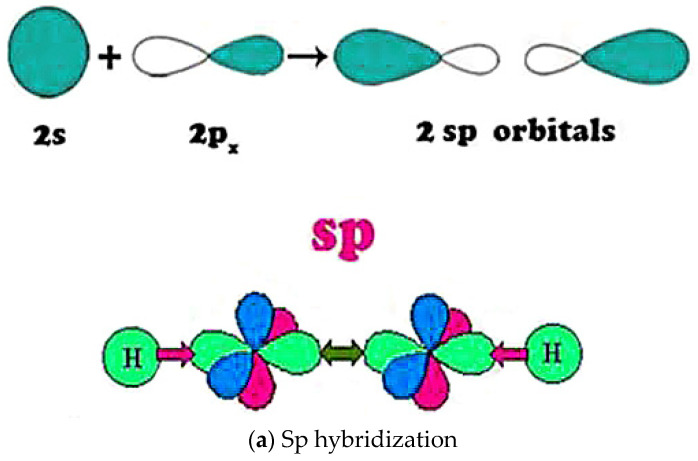
Bonding mode of sp and sp^2^.

**Figure 9 molecules-27-07867-f009:**
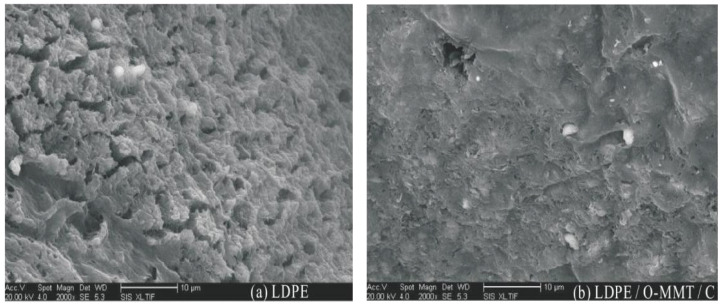
SEM photographs of two breakdown samples.

**Figure 10 molecules-27-07867-f010:**
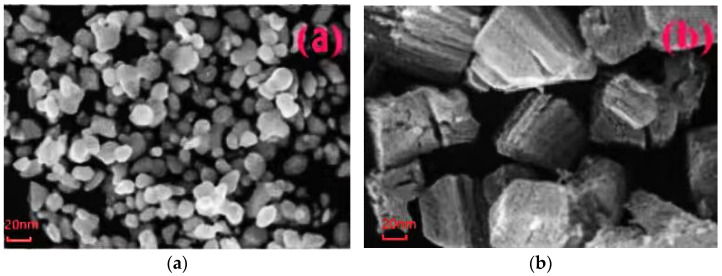
TEM photographs of PS and nano-ZnO/PS. (**a**) TEM photograph of PS; (**b**) TEM photograph of nano-ZnO/PS.

**Figure 11 molecules-27-07867-f011:**
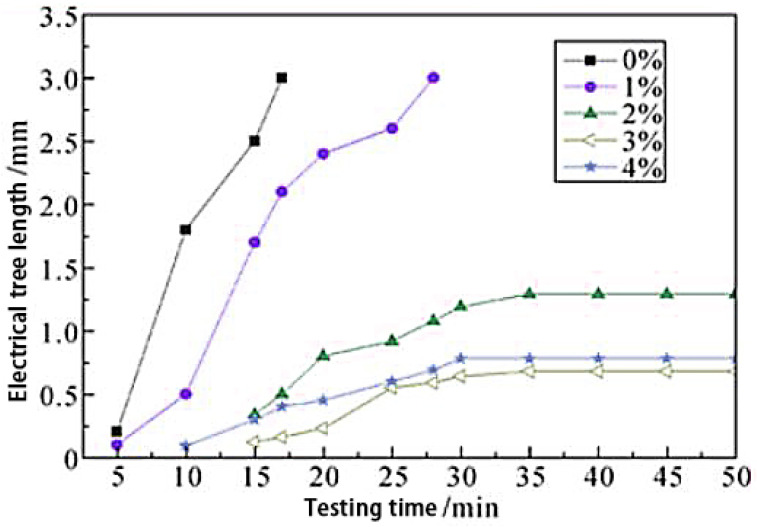
Electrical tree growth curves of nano-ZnO/LDPE with different ZnO content.

**Figure 12 molecules-27-07867-f012:**
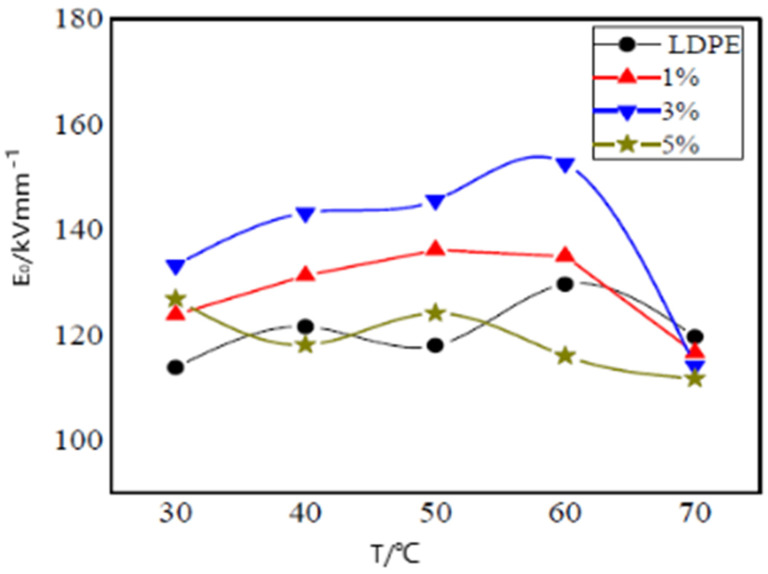
Weibull breakdown field strength temperature curves of different composites.

**Figure 13 molecules-27-07867-f013:**
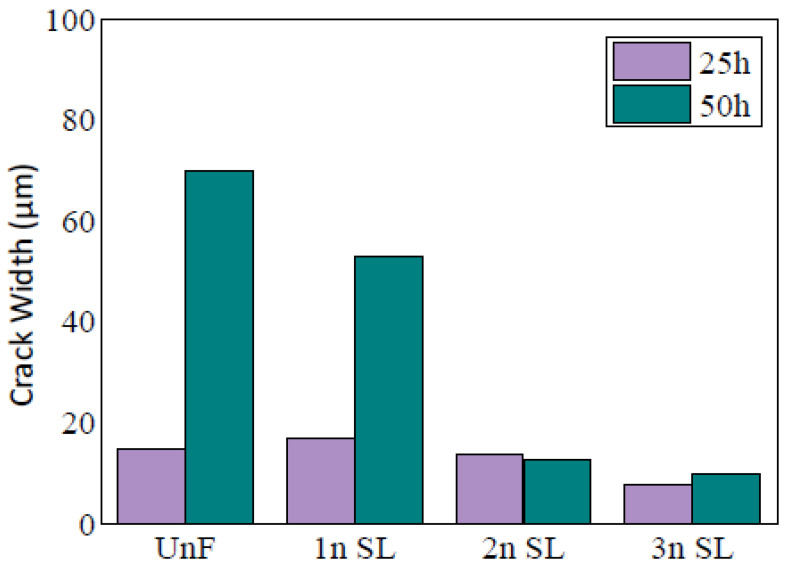
The widths of the electric cracks in the different samples.

**Figure 14 molecules-27-07867-f014:**
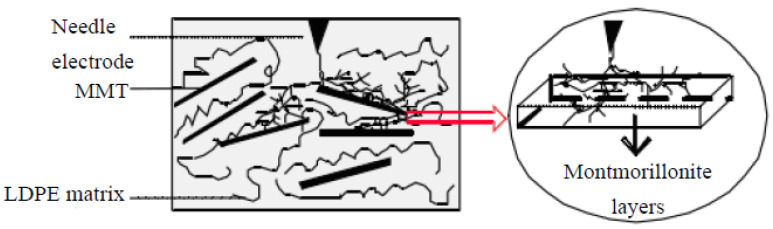
Effect of nano-MMT particles on the electrical tree growth channel.

**Figure 15 molecules-27-07867-f015:**
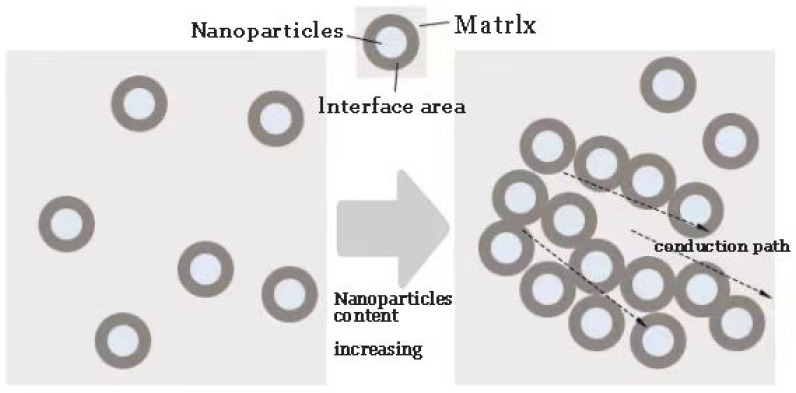
Charges conduction path formation of nano-polymer.

**Figure 16 molecules-27-07867-f016:**
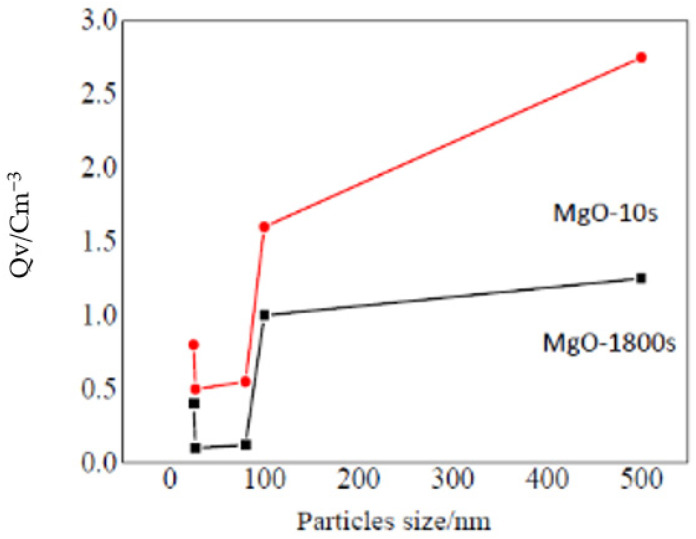
Relation between medium charge density and particle size of nano-MgO.

**Figure 17 molecules-27-07867-f017:**
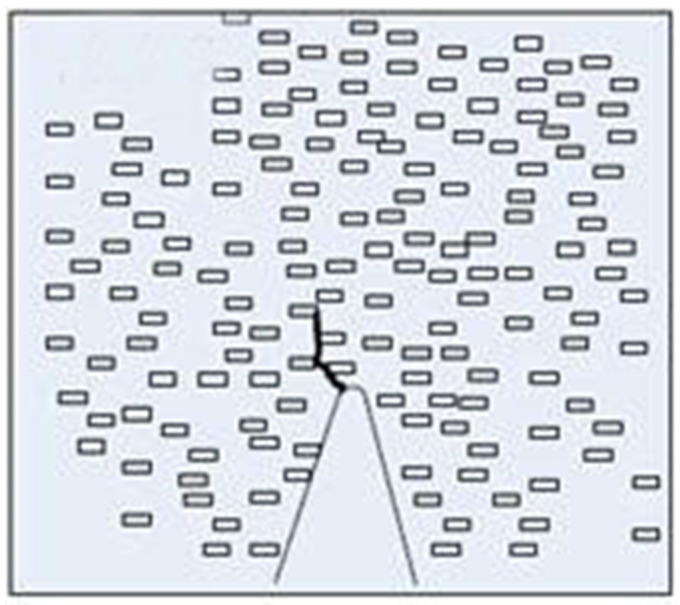
Breakdown path of nanoMMT/LDPE.

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
