# Peer review of "Research Progress of Polymers/Inorganic Nanocomposite Electrical Insulating Materials"

_molecules, 2022, doi:10.3390/molecules27227867_

Round 1

Reviewer 1 Report

This review addresses the research of dielectric polymer/inorganic nanocomposites. It covers the background, theories, and research progress. The logic flow is clear, but more information should be provided to make the review comprehensive. A major revision is needed before acceptance.

1.      Dielectric nanocomposites can be used in different applications with different focuses. Which research area is this review mainly focusing on? I feel it might be electrical insulation. If yes, the title and the abstract should be revised.

2.      The description of the theories is too simple. Equations and references should be added.

3.      In Figure 1, what does the “high polymers matrix” mean?

4.      There are too few figures in section 4. And those figures did not show the key performance of the dielectric nanocomposites.

5.      In Section 5, no outlook and perspective is provided.

6.      There are too few references. And most of the references are not from top-level materials journals.

Reviewer 2 Report

Dear authors:

The introduction section needs to be revised. (1) I suggest the authors to add a paragraph of literature review summarizing some commonly used nanoparticles in PEI matrix and their fabrication methods. (2) In addition, the novelty and advance of this work should be highlighted and illustrated. (3) some papers can be cited in the first paragraph regarding the use of polymer nanocomposites for different applications: Core/Shell Conjugated Polymer/Quantum Dot Composite Nanofibers through Orthogonal Non-Covalent Interactions." Polymers 8.12 (2016): 408.; Structural and mechanical properties of polymer nanocomposites." Materials Science and Engineering: R: Reports 53.3-4 (2006): 73-197.; "Device applications of polymer-nanocomposites." Biopolymers· PVA hydrogels, anionic polymerisation nanocomposites (2000): 163-205.

In line no. 63, the sentence: "According to statistics, a 60% power system failure originates from electrical insulation issues.", please show references to this claim. As it is shown e.g. in: (DOI: 10.1109/ACCESS.2021.3069144 or DOI: 10.3390/polym13030375)

In line no. 69, the sentence: "Therefore, the dielectric properties of polymer nanocomposites are superior to those of traditional polymer composites", please showed the relevant references.

Section 2 with the name: 2. Fundamental effects of nanoparticles contain only 5 references, which is very little.

In the section 2 please describe a number of different processes (such as the a-, b- and g-relaxations and the Maxwell-Wagner-Sillars (MWS) effect) in an AC electric field, the intermediate dipolar effect (IDE), and DC conductivity were described in [DOI: 10.3390/polym14112202] relation to polarisation mechanisms.

Please describe also Tanaka model in section 2 (Tanaka, T.; Kozako, M.; Fuse, N.; Ohki, Y. Proposal of a multi-core model for polymer nanocomposite dielectrics. IEEE Trans.

Dielectr. Electr. Insul. 2005, 12, 669–681.)

About adding nano-MgO particles on polymer Matrix use a next relevaation references (DOI: 10.3390/ijms222312752)

Reviewer 3 Report

This paper reviewed the recent progress of polymers/inorganic nanocomposite dielectric materials. However, I can not recommend it publish due to the following reasons.

1\There are many literatures about dielectric nanocomposites. However, the authors have not given a comprehensive review. That is, this review did not contain enough information about recent progress of polymers/inorganic nanocomposite dielectric materials.

2\The caption of Figure 6 “Single-layer structure model” should be revised.

3\Fundamental effects of nanoparticles “(1)Macroscopic quantum tunneling effect, (2)Small-size effect, (3)Surface effect, (4)Quantum size effect” are well known and they should be removed.

4\ Abbreviated terms should be defined, such as “PI-AI system and PI-SI system”,” K0”

5\Page 9-11, the list of Professor is very stiff, it should be revised.

Round 2

Reviewer 1 Report

The authors have addressed all my questions.

Reviewer 2 Report

Dear authors.

There has been a significant change in the submitted and revised article that will improve the quality of the presented article. However, I would like to ask you to incorporate my other comments that may increase the attractiveness of the submitted article.

1. What does mean the acronym "PEI" in the line 40 (...nanoparticles modified by PEI possess high electron density...)

2. Your statement in line 61 and 62: "According to statistics, a 60% power system failure originates from electrical insulation issues" does not correspond to reference 8. The relevant link is in the articles (DOI: 10.1109/ACCESS.2021.3069144 and DOI: 10.3390/polym13030375). Please use these references directly, or other corresponding relevant references.

3. Line 74 - 77. The baseic literatures in this area are also: Singha, S.; Thomas, J.M. Dielectric Properties of Epoxy Nanocomposites. IEEE Trans. Dielectr. Electr. Insul. 2008 and Kochetov, R. Thermal and Electrical Properties of Nanocomposites, Including Material Processing. Ph.D. Thesis, Wöhrmann Print Service, Zutphen, The Netherlands, 2012.

4. Macroscopic quantum tunneling effect was firstly described by Fritz Wolfgang London, please use relevant reference

5. Please describe also thermal activation energy in section 2. Fundamental effects of nanoparticles

6. Figure. 3 is weakly readability

7. In a line 361 please use a description from literature: Tanaka, T.; Montanari, G.C.; Malhaupt, R. Process, understanding and challenges in the field of nanodielectrics. IEEE Trans. Dielectr. Electr. Insul. 2004, 11 and Tanaka, T.; Kozako, M.; Fuse, N.; Ohki, Y. Proposal of a multi-core model for polymer nanocomposite dielectrics. IEEE Trans. Dielectr. Electr. Insul. 2005, 12.

8.  Line: 409: Not prof. Hornak, but Dr. Hornak  and not only 2022 but in also previous years

9. Fig. 9: It is possible use to a picture from TEM?

10. In Conclusion please write also other authors who studied the effect of some types of nanoparticles on dielectric response in polymer materials.

Reviewer 3 Report

This review does not contain comprehensive literature involving polymers/inorganic nanocomposite electrical insulating materials. I suggest the authors supply more references to fully show the progress in this field. 
